# Density, Temperature, and Comingled Species Affect Fitness within Carrion Communities: Coexistence in *Phormia regina* and *Lucilia sericata* (Diptera: Calliphoridae)

**DOI:** 10.3390/insects14020139

**Published:** 2023-01-29

**Authors:** Patricia Okpara, Sherah VanLaerhoven

**Affiliations:** Department of Integrative Biology, University of Windsor, 401 Sunset Ave, Windsor, ON N9B 3P4, Canada

**Keywords:** medico-legal entomology, forensic entomology, species interactions, sub-lethal effects, facilitation, competition, heterospecific assemblages

## Abstract

**Simple Summary:**

Blow fly life cycles are commonly used to estimate the time elapsed since death of unattended death when this interval is greater than 48 h. Female flies lay eggs in clusters, resulting in larval masses that vary in density and species, which may impact resulting adult fly size, reproductive ability, and even survival to adult. How multiple blow fly species all use the same limited dead animal resources without causing extinction of one or more species is still poorly understood. Therefore, the goal of this research was to test the role of temperature, number of individuals, and presence of other species during immature development in adult fly size and survival to adult between two common blow flies. The number of individuals and temperatures experienced by the larvae during development was manipulated in the presence of individuals of the same or different species in the laboratory. Survival to adult and body size was larger for one species (*Phormia regina*) when developing with the other (*Lucilia sericata*). In contrast, the other species’ (*L. sericata*) survival depended primarily on temperature. However, body size increased with the presence of the second species (*P. regina*), depending on temperature and number of individuals. The results of this study suggest that while temperature may be the most critical factor influencing interactions between blow fly species, the number of individuals and the ratio of individuals of each species must be considered when using these data for time of death estimates in forensic entomology. Given the critical role of temperature in influencing these outcomes, climate change has the potential to disrupt the ability of these species to coexist in their current distributions.

**Abstract:**

Blow fly (Diptera: Calliphoridae) interactions vary between competition and facilitation. Female blow flies engage in aggregated egg-laying, resulting in larval feeding masses differing in density and species composition. Numerous species are abundant within the same season, and some oviposit near or directly on eggs of other species, modifying their oviposition location choice depending on the presence or absence of other species. The ability to coexist on carrion, a temporary resource, was successfully attributed to resource, spatial, and temporal heterogeneity. Despite these broad categorizations, the specific mechanisms of coexistence within blow fly communities require further investigation. This study investigates variation in temperature and larval density as potential mechanisms of coexistence between two forensically important blow fly species: *Lucilia sericata* Meigen and *Phormia regina* Meigen (Diptera: Calliphoridae). Larval density, species ratio mix, and ambient temperature during development were manipulated in the presence of conspecifics and heterospecifics in the laboratory, and the fitness of each species was measured. In heterospecific treatments, the survival and body size of *P. regina* increased even at high ambient temperatures. In contrast, the survival of *L. sericata* remained unaffected by density or presence of heterospecifics, whereas body size increased in *L. sericata*-dominated heterospecific treatments depending on temperature and density. The negative effects of density were observed at high ambient temperatures, suggesting that density impacts are a function of ambient temperature. Overall, species coexistence was dependent on temperature, which mediated the outcome of species interactions.

## 1. Introduction

Carrion-based communities are made up of species interaction networks [1]. They are classified on a continuum of interactive to non-interactive relationships depending on the availability of space on carrion resources. In non-interactive communities, large amounts of unused space are created by reduced populations resulting from changes in abiotic environments [2]. On the other hand, interactive carrion-based communities consist of saturated resource space, resulting in strong species interactions between individuals, such as competition [3], predation [4], as well as mutualism and facilitation [5], that can change due to presence or absence of a species and the quality of the shared resources.

Necrophagous species’ ability to successfully coexist on carrion, a transient resource, is attributed to resource, spatial, or temporal heterogeneity. Spatial resource heterogeneity describes uneven and diverse microhabitats within a carrion resource, which offers different growth conditions due to its patchiness and supports increased species diversity. This positive relationship between spatial environmental heterogeneity and species diversity is supported by the example of higher pest species diversity observed in host plants with greater size and morphological complexity [6]. Spatial heterogeneity proposes that species occupy different areas of shared resources to reduce levels of interspecific competition, such as was found when increases in spatial heterogeneity across landscapes increased avian species richness over time [7] or with a spatially heterogeneous resource that supported higher algal species [8].

Temporal heterogeneity describes species separation in time when utilizing shared resources; that is, species use daily and seasonal patterns of colonization to exploit resources and reduce competition levels [9,10]. Ody et al., 2017 [10] observed *Calliphora vicina* Robineau-Desvoidy (Diptera: Calliphoridae) ovipositing at temperatures as low as 10 °C but not at 40 °C, and *Lucilia sericata* Meigen (Diptera: Calliphoridae) ovipositing at temperatures as high as 40 °C but not at 10 °C in conspecific treatments. Despite these mechanisms promoting species coexistence, several blow fly species are abundant within the same seasons, and some oviposit near or directly on eggs of other species [11].

Insects cannot regulate their body temperature and rely on environmental conditions to regulate their metabolic and developmental processes. Therefore, temperature can affect the strength of the interactions during insect development and the competitive abilities of the species within the same trophic level [11,12]. Populations of *Phormia regina* Meigen (Diptera: Calliphoridae) and *L. sericata* or *C. vicina* grown together at two different temperatures of 20 °C and 25 °C showed that at 20 °C, *P. regina* gained mass more rapidly in the presence of *L. sericata* as did *L. sericata* in the presence of *P. regina* [11]. However, when grown at 25 °C, *P. regina* larvae gained mass more slowly in the presence of *L. sericata* or *C. vicina* than when with conspecifics [11]. Thus, not only do environmental conditions, such as temperature, have the potential to influence species interactions, but also the outcome of these interactions.

Insect development occurs within a thermal range with minimum (T_min_) and maximum limits (T_max_) that represent the lowest and highest temperature at which insect development can occur; in between these are tolerance zones and an optimal range [13]. Several studies reported a decrease in survival when species develop in temperatures higher than the species optimal range, including *Aedes aegypti* Linnaeus (Diptera: Culicidae) [14], *Ephestia cautella* Walker (Lepidoptera: Pyralidae) [15], *Plodia interpunctella* Hὒbner (Lepidoptera: Pyralidae) [15], and *Lucilia sericata* [16]. Yet temperature effects on insect development can differ between conspecific and heterospecific populations [17]. Davis et al., 1998 [12] studied the effects of temperature on three species’ interactions: *Drosophila melanogaster* Meigen, *Drosophila simulans* Sturtevant, and *Drosophila subobscura* Collin (Diptera: Drosophilidae). Mixed culture interactions resulted in both having higher survival and larger body size and a shift in the optimal thermal range of *D. melanogaster* and *D. subobscura* [12]. Temperature can alter the metabolic rates and competitive ability of different species and mediate their interactions.

In addition to temperature, larval density can negatively affect the life history traits of insect populations that grow on limited resources [18]. This includes reduced adult body size, which also means decreased fecundity [19,20], reduced survival [21], and increased developmental times [22]. Larval density mediated the outcome of interspecific competition between two blow fly species, *L. sericata* and *C. vicina* [23]. The effects of interspecific competition were greater at densities greater than ten larvae/g of liver for *L. sericata*. However, the opposite was true for *C. vicina* with greater effects of intraspecific than interspecific competition [23]. Density mediated the outcome of the interactions by changing the intensity experienced by the individuals present. Increasing larval density should be expected to increase the intensity of the species interaction experienced by individuals.

Given that carrion is a patchy and ephemeral resource, gravid females should maximize their offspring’s fitness by choosing to oviposit in areas that limit high levels of interspecific competition experienced by their offspring [24]. Females oviposit in aggregates, presumably to help reduce desiccation experienced by the eggs and promote larval development by shared digestive enzymes that facilitate decomposition and inhibit competitive microbes [25,26]. This aggregated oviposition behaviour results in unsaturated niche space for other blow fly species to colonize, thereby supporting coexistence over a shared resource. Similarly, this should allow highly competitive species to coexist on a limited, shared resource [27]. Within a single carrion, the egg aggregates may vary in location, density, and species composition. Additionally, it is possible that initial densities of each species present might influence the outcome of species interactions as they have more opportunities to dominate the resource. This phenomenon is known as founder control, it assumes that the first arriving species will have the greatest opportunity to dominate a resource, thereby giving it a competitive advantage over other species present [28].

*Lucilia sericata* and *P regina* are common members of the carrion community, native to Canada, and colonize the same carrion resource [29,30]. *Lucilia sericata* is known as the sheep blow fly or the green bottle blow fly, and it develops at a thermal range of 10 °C to 32.5 °C [17,31]. It is considered a weak interspecific competitor, exhibiting decreased survival in mixed-species treatments [23] and ovipositing without delay as a strategy to avoid potential competition [32]. On the other hand, *P. regina,* known as the black blow fly, has a thermal range of 10 °C to 40 °C [33]. There is limited investigation of possible species interactions between these two species, despite their common comingled assemblages [11]. This research examined how temperature and density influence species interactions, and therefore coexistence, within conspecific and heterospecific larval communities of *L. sericata* and *P. regina*. We measured survival and adult body size under different temperature and density treatments. Given the developmental temperature range for *L. sericata* and *P. regina* [31,33], we predicted that survival would increase with increasing temperatures; however, this would be a function of species thermal range, with survival being the lowest at the upper thermal limits of each species. As a result, *P. regina* would have higher survival than *L. sericata* at high developmental temperatures in both hetero and conspecific communities. Following results from Hans and VanLaerhoven [11], we predicted that survival of *P. regina* and *L. sericata* will remain unchanged in the presence of heterospecifics when compared to conspecifics within the densities and temperatures tested herein. As the effects of density on insect development have were extensively studied [20,21], we predicted a negative relationship between density and fitness (survival and body size), as carrion is a limited resource and increasing densities are expected to increase levels of intraspecific and interspecific competition in conspecific and hetero-specific communities, respectively. We also predicted an interaction between density and temperature such that the negative effects of density would be greatest at temperatures above each species’ thermal threshold in both conspecific and heterospecific populations.

## 2. Materials and Methods

### 2.1. Colony Maintenance and Larvae Collection

Colonies of adult *L. sericata* and *P. regina* were reared and maintained at 25 ± 0.2 °C, here and below mean ± S.E., are given on a 12D:12L cycle in aluminium mesh screen cages (46 × 46 × 46 cm) (BioQuip Products, Item ID: 1450C) at the University of Windsor, Windsor, Ontario, Canada. The colonies were established from wild individuals collected in 2005 using the king wasp traps (King home and garden products, Item ID: 56789) lined with paper towels and using the liver as an attractant. Wild flies were caught and added to the colonies yearly to supplement colonies and reduce any possible effects of inbreeding depression. Colonies were provided with water in Erlenmeyer flasks with dental wicks to prevent drowning, sugar cubes, and protein powder *ad libitum* in the form of a paste as a carbohydrate and protein source, respectively [34]. Colonies were provided with liver, which served as a rich protein source for females to ensure complete ovarian development. Egg masses of approximately 500 eggs were collected using 50 g of pork liver as an oviposition medium for 24 h to establish experimental cages of each species. Once adults were mature and gravid within the experimental cages, eggs were obtained for the experiment by providing liver for four hours. Egg masses collected were checked every four to six hours for eclosion. Egg masses from each species hatched at 18.8 ± 0.8 h for *P. regina* and 21 ± 2.5 h for *L. sericata* at a temperature of 25 ± 0.2 °C on a 12D:12L cycle. Less than four- to six-hour old larvae were transferred onto 50 g of pork liver using a paintbrush and then placed directly on aspen wood shavings (top bedding, Item ID: large wood shavings), filled to one-third of the volume into 1 L Bernardin mason jars. Wood shavings served as a pupation medium for the blow flies. Jars were covered with landscape tarp (Quest Brands Inc. ID: WBS 50) to allow ventilation and a metal ring lid to prevent maggots from escaping.

### 2.2. Experimental Design

Treatments were a full factorial design with three developmental temperatures of 15 °C, 25 °C, and 35 °C, five larval densities of 25, 50, 100, 200, and 400 larvae per 50 g of pork liver, and five species treatments of 100% *P. regina*, 75% *P. regina* to 25% *L. sericata,* an equal 50:50 ratio, 25% *P. regina* to 75% *L. sericata,* and 100% *L. sericata*. The equal 50:50 ratio was created at a larval density of 25 by setting up five jars with 12 *L. sericata*: 13 *P. regina*, and five jars with 13 *L. sericata*: 12 *P. regina.* Each density and species treatment was replicated ten times within each temperature treatment. Experimental jars were placed in a growth chamber (Conviron Adaptis A1000, Winnipeg, MB, Canada) programmed with a photoperiod of 12L:12D, 70% R.H., and the appropriate temperature treatment. For each temperature treatment, three growth chambers were programmed at the specific temperature, and verified by a datalogger (Smart button, ACR Systems Inc., Surrey, BC, Canada). Within each growth chamber, species ratio treatments were randomly assigned one of five shelves, with density jar treatment placed randomly within the shelf. Based on preliminary data at the highest density of 400 larvae, no increase in temperature due to maggot masses was observed. At the highest density, 50 g of liver is sufficient for 100% survival, requiring no additional liver. When the adult flies emerged in the rearing jars, they were killed and counted to record each species’ percent survival (adult emergence) at each density, species, and temperature treatment. The length of the female posterior cross-vein to dm-cu vein [24] of the left-wing was measured to represent adult size, using a micrometer in a microscope.

### 2.3. Statistical Analyses

All analyses were completed in JMP (SAS Institute Inc, Version 16.1.0 (539038)). A general linear model (GLM) was used to examine the effect of density, temperature, species ratio, and the interactions of these variables on the adult body size of both species. Survival was analyzed using a three-way ANOVA to examine the effects of temperature, density, species ratio, and the interaction between these variables on the survival of both species. Normality was tested using the Shapiro–Wilks test, and homogeneity of variance was tested using Bartlett’s test. A significant level was set as α = 0.05.

## 3. Results

### 3.1. Body Size

#### 3.1.1. *Phormia Regina*

Adult body size was influenced by an interaction between species treatment, density, and temperature (F_3,997_ = 3.9 *p* = 0.0079) (Table 1, Figure 1). In single species, or 100% *P. regina* treatment, the largest adults were observed at 25 °C in lower larval densities, followed by adults at 15 °C in higher larval densities, and then adults at 35 °C with lower larval densities. The smallest adults were observed at higher densities at 35 °C.

Compared to *P. regina* alone (100% single species), across all temperatures and across all densities, adult body size was larger in heterospecific treatments except at density of 25 at 15 °C, and densities of 25 and 50 at 25 °C. Despite still being larger than *P. regina* alone, when comparing body size between different heterospecific species ratio treatments, the *L. sericata* dominant ratio (25:75) exhibited opposing trends across density occurring at 25 °C, with a decrease in body size with an increase in density, compared to the other heterospecific species ratios (Figure 1).

#### 3.1.2. *Lucilia Sericata*

Adult body size was influenced by an interaction between species treatment, density, and temperature (F_3,997_ = 8.54, *p <* 0.001) (Table 1, Figure 2). In single species, or 100% *Lucilia sericata*, there was no effect of density at 15 °C and 35 °C, where adults were larger overall at 15 °C than 35 °C. However, at 25 °C, adults were largest at densities of 100 and lower, but decreased in body size as density increased such that adults did not differ from 15 °C at densities of 200, and smaller than 15 °C at densities of 400.

Compared to *L. sericata* alone (100% single species), adult body size was larger, regardless of temperatures or species ratio, except at 25 °C for *L. sericata* dominant species ratio (25:75) at the low density of 50 larvae (Figure 2b) and at 35 °C at higher densities. Despite still being larger than *L. sericata* alone, when comparing body size between different heterospecific species ratio treatments at 35 °C, increasing density resulted in decreasing body size at equal species ratios (50:50) and *P. regina* dominant ratios (75:25) (Figure 2c).

### 3.2. Survival

#### 3.2.1. *Phormia Regina*

The effect of temperature, density, and species treatment on the adult survival of *Phormia regina* was analyzed using analysis of variance. Survival was influenced by an interaction between species treatment, density, and temperature (F_3,584_ = 6.02, *p =* 0.0005) (Table 2, Figure 3). In single species 100% *P. regina* treatments, survival was highest at 25 °C and lower at 15 °C across all densities (Figure 3). At 35 °C, survival was equivalent to that at 25 °C at the lowest densities, declining as density increased to below survival at 15 °C at the highest density.

Compared to *P. regina* alone (100% single species), survival increased with temperature for all heterospecific treatments, regardless of the ratio between species; except for the *P. regina* dominate ratio at 25 °C for densities 25 and 200, survival was higher than at 35 °C (Figure 3). At densities of 100 and higher, survival was greater in heterospecific treatments than the single species *P. regina* treatment, with the exception of equal ratio (50:50) and *P. regina* dominant (75:25) at a density of 200. Between heterospecific treatment ratios within a single temperature, survival of *P. regina* either stayed the same or increased with density. Between heterospecific treatment ratios within a single density, survival of *P. regina* increased with temperature (Figure 3).

#### 3.2.2. *Lucilia Sericata*

Survival was influenced by an interaction between density and species ratio treatments (F_3,584_ = 7.19, *p <* 0.0001), but temperature did not interact with density or species ratio ((F_3,584_ = 0.77, *p =* 0.5096) (Table 2, Figure 4). As temperature increased, survival of *L. sericata* decreased. In single species 100% *L. sericata* treatments, density did not affect survival. Compared to *L. sericata* alone, survival in heterospecific treatments increased with increasing density, although at no point did it differ from overall survival of *L. sericata* treatments alone (Figure 4).

## 4. Discussion

Carrion can be described as an interactive community, as it is often saturated with different carrion insects that exhibit strong interactions between individuals within or between trophic levels [2]. These strong species interactions can be influenced by temperature and density [18,24]. However, few studies investigated the influence of temperature and density together, or with species interactions within blow fly communities, to evaluate effects on fitness.

The positive effects of *L. sericata* on the survival of *P. regina* at the highest temperature tested were unexpected, as we observed decreased survival of *P. regina* in single-species treatments with increasing larval density due to increased levels of competition on a limited resource [3,18,19,20,21]. These positive effects could be due to decreased interspecific interactions at this temperature, as 35 °C exceeds the temperature threshold for *L. sericata*. At this high temperature, there is reduced survival of *L. sericata* as development progresses, thereby decreasing the overall density within a jar, potentially reducing inter-specific interactions experienced by *P. regina.* Additionally, the specific density of *P. regina* is lower after the death of *L. sericata* larvae compared to the density of *P. regina* in the single-species cultures. Thus, improved survival of *P. regina* at this temperature could be a combination of benefits from the enzymatic secretions of *L. sericata* that might aid in breaking down resources and inhibit the growth of harmful microbes, and also increased resource availability due to the death of *L. sericata* larvae. Overall, *P. regina* in the presence of *L. sericata* resulted in a shift in the optimal developmental temperature range of *P. regina* from 25 °C in single-species culture to 35 °C in mixed-species culture.

Greenberg and Kunich [35] suggested a possible mechanism to explain the coexistence of different species on a limited resource; larvae benefit from larval aggregations by multiple individuals penetrating tissues and releasing enzymatic secretions that aid digestions. However, these might not be the only positive effects experienced by larvae within multi-species larval aggregates. We speculate that antimicrobial properties of the larval sections of *L. sericata* might contribute to the increased survival of *P. regina* by acting against competing microbes and bacteria, such as *Staphylococcus aureus* Rosenbach and *Staphylococcus epidermis* Evans (Bacillales: Staphylococcaceae) [36]. This reduces overall competition that might affect the development and survival of *P. regina* [36]. Both bacteria species are part of the normal flora, commonly found on the skin, hair, nose, and respiratory tract of organisms such as humans and animals, and are likely present at even higher concentrations after death [36]. 

Previous studies suggested that *L. sericata* is a weak interspecific competitor, displaying low survival during development with other blow fly species [23,37,38]. However, one study suggests that *L. sericata* behaves differently based on geographic origin [39]. Martinez-Sanchez et al., 2007 [39] found that Spanish populations of *L. sericata* showed greater levels of mortality than United Kingdom populations when reared at similar densities. While the opposite was true for body size, the Spanish populations were significantly larger than the UK populations. The underlying mechanism for these observations remains unclear, but they demonstrate that the geographic origin of an insect population contributes to differences in its competitive ability. The current data suggest that the survival of *L. sericata* is unchanged when developing with *P. regina,* but increases with larval density. We suspect that while there are no positive influences on the survival of *L. sericata* by *P. regina*, perhaps *L. sericata* is a better interspecific competitor than previously assumed, and *L. sericata* benefits from the decreased intra-specific interactions, as species densities in mixed cultures are less than single-species cultures.

Atkinson and Sibly 1997 [40] proposed the temperature–size rule, which states that an individual’s adult body size is a product of environmental conditions, such as temperature. Thereby, individuals developing at low temperatures exhibit larger adult body size [40]. The influence of temperature on the adult body size of *L. sericata* was evident within all temperature treatments, and the largest females were observed at 25 °C when compared to other temperatures. This discrepancy with the temperature–size rule can be explained by limited resource availability and energy allocation, prioritizing faster development over larger adult body size. At lower temperatures, individuals have a longer development time, the rate of decomposition is lower, and the resource is not utilized as rapidly. For individuals to achieve a significant size, they must postpone development. Longer development increases their risk of predation and potentially increased competition with other carrion insects as they continue to colonize the resource.

Overall, female body size of *P. regina* and *L. sericata* increased in the presence of heterospecifics than conspecifics at all temperatures tested. This could mean that the enzymatic and antibiotic secretions of *L. sericata* might prevent microbial growth and aid in the breakdown of food resources, making it more readily available and easy to utilize. Females can use this opportunity to allocate more resources to their body size, thereby increasing their fecundity. Given that the survival of *L. sericata* was not affected by the presence of *P. regina*, it was interesting to measure larger adult body size of *L. sericata* during development with *P. regina*. This could suggest that *L. sericata* benefits from decreased intra-specific interactions while successfully utilizing the resource through the breakdown by enzymatic secretions.

*Phormia regina* and *L. sericata* were reported to inhabit and feed on the same carrion [29,30]. They overlap in their distribution in both space and time, meaning they often interact with one another, and the outcomes of these interactions can influence their development and fitness negatively or positively. We incorporated different species ratios to test an aspect of the ‘founder control’ phenomenon, which assumes that the first arriving species, usually the more abundant species, will have the greatest opportunity to dominate a resource, thereby giving it a competitive advantage over other species present [28]. From our study, we suggest that the possible effects of ‘founder control’ will be dependent on a species thermal range, that is, within a favorable range, they can exert competitive advantage when present in higher proportions than other species. However, beyond these favourable conditions, individuals have no competitive advantage, regardless of the proportions of each species available. While the present study does not incorporate temporal priority effects, it provides useful information on the possible mechanisms that support species coexistence on a limited resource.

This study utilized constant temperatures in a laboratory environment. Fluctuating temperatures are likely to impact outcomes of species interactions in blow flies, as suggested by the differential effects of temperature on development rate and weight gain of single species versus comingled heterospecific assemblages of blow flies [11] and the effect of fluctuating temperature on increasing the development rate of blow flies [41,42,43]. However, to date, the combined effects of fluctuating temperatures and heterospecific assemblages on the development rate and survival of blow flies were not studied.

This study’s results are particularly important, as *P. regina* and *L. sericata* are carrion flies that arrive within minutes after death and are reliable forms of calculating the post-mortem interval (PMI) for homicide and suicide victims [28]. It is important to investigate biotic and abiotic factors that can influence colonization and development, as forensic entomologists rely on predictable patterns and behaviours during forensic investigations [44]. By understanding these specific mechanisms, and the exploring factors that influence species coexistence within carrion communities, we gain critical understanding to aid in the defining the predictability of species utilizing carrion resources that are vital in forensic entomology, important decomposers for the essential functioning of our ecosystems, and glimpses into potential effects of climate change on these species.

## Figures and Tables

**Figure 1 insects-14-00139-f001:**
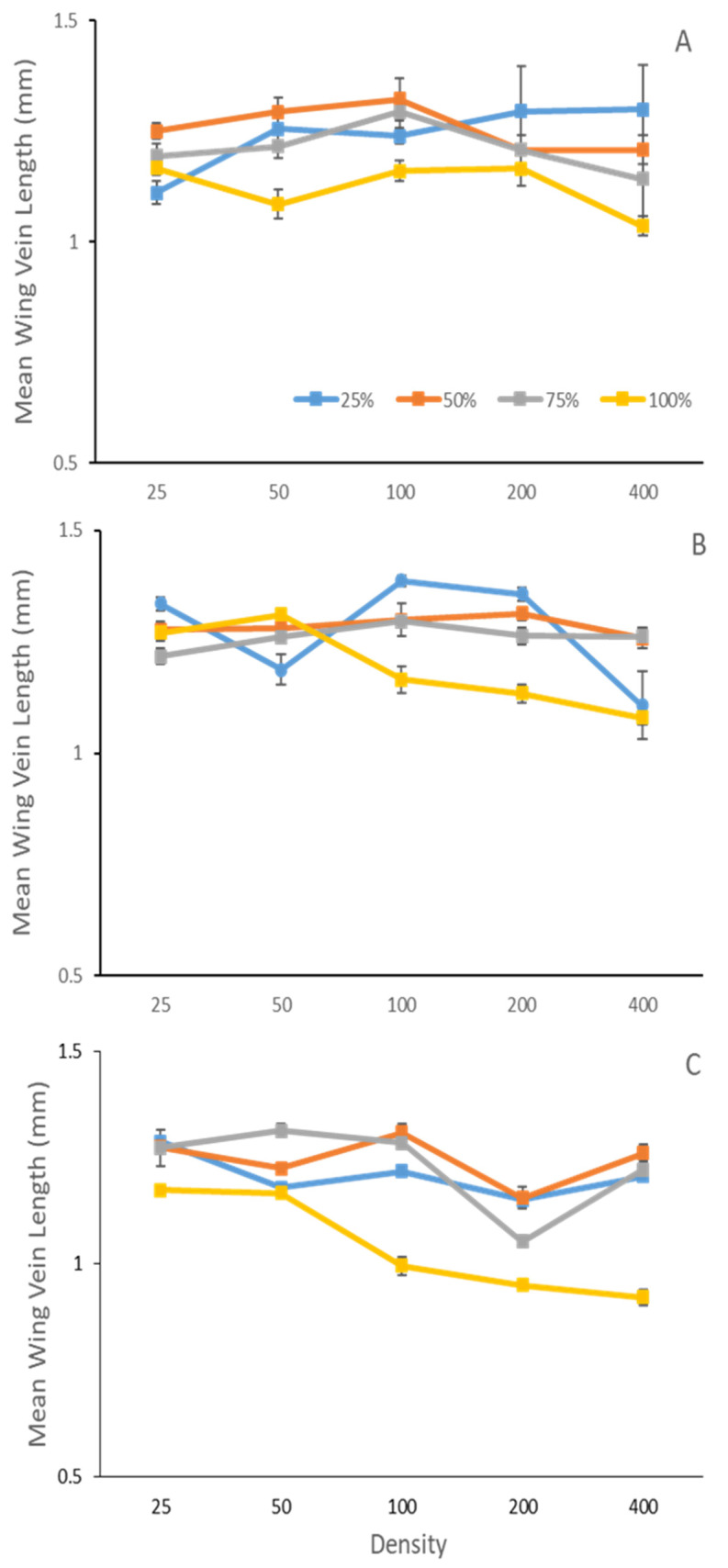
Mean ± SE adult body size of *Phormia regina* Meigen (Diptera: Calliphoridae) (as measured by the length of the posterior cross-vein to dm-cu vein of the left wing) reared within larval ratios of 100% *P. regina*, 75% *P. regina* to 25% *Lucilia sericata* Meigen (Diptera: Calliphoridae), equal 50:50 ratio, or 25% *P. regina* to 75% *L. sericata* at (**A**) 15 °C, (**B**) 25 °C, and (**C**) 35 °C across densities (larvae/jar). There was an interaction between density, species treatment, and temperature (F_3,997_ = 3.9 *p* = 0.008) between 15 °C and 25 °C, such that at 15 °C *P. regina* body size was smaller than other species ratios at low density and larger than other species ratios at high density. The opposite trend occurred at 25 °C, with a decrease in body size with an increase in density, compared to the other heterospecific species ratios (Figure 1).

**Figure 2 insects-14-00139-f002:**
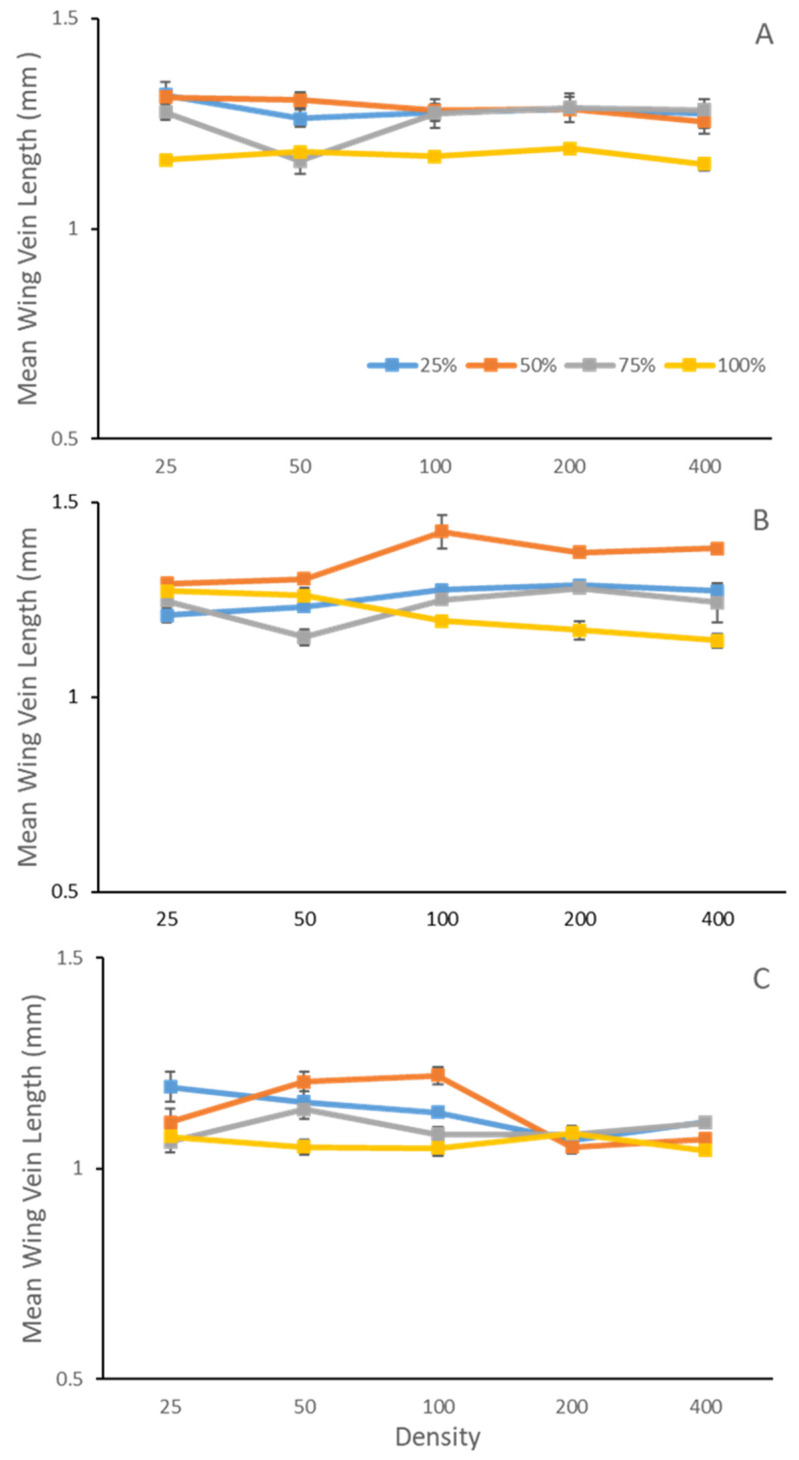
Mean ± SE adult body size of *Lucilia sericata* Meigen (Diptera: Calliphoridae) (as measured by the length of the posterior cross-vein to dm-cu vein of the left wing) reared within larval ratios of 100% *L. sericata*, 75% *L. sericata* to 25% *Phormia regina* Meigen (Diptera: Calliphoridae), equal 50:50 ratio, or 25% *L. seriata* to 75% *P. regina* at (**A**) 15 °C, (**B**) 25 °C, and (**C**) 35 °C across densities (larvae/jar). Adult body size was influenced by an interaction between species treatment, density, and temperature (F_3,997_ = 8.54, *p <* 0.001).

**Figure 3 insects-14-00139-f003:**
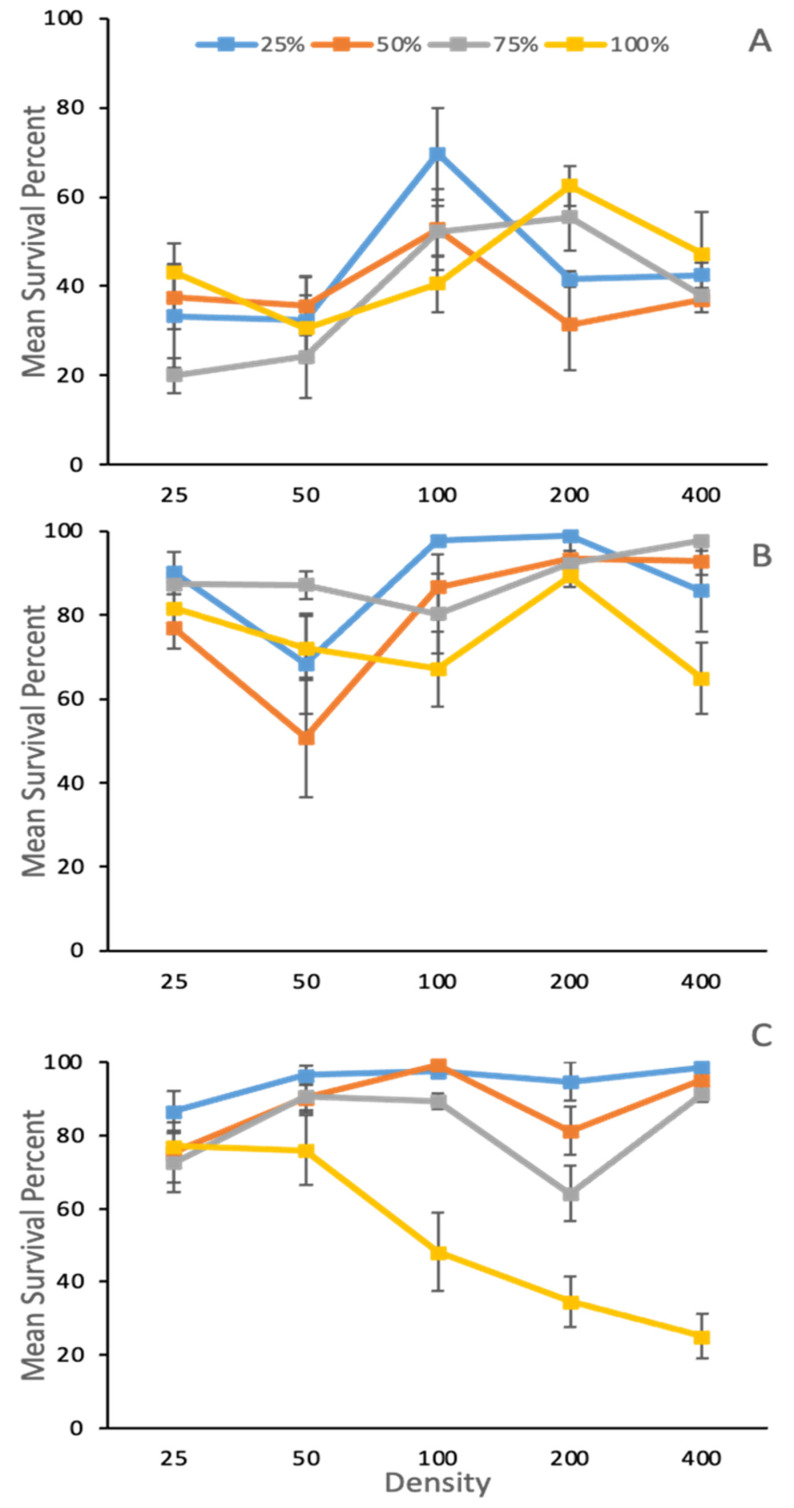
Mean ± SE percent survival to adult of *Phormia regina* Meigen (Diptera: Calliphoridae) reared within larval ratios of 100% *P. regina*, 75% *P. regina* to 25% *Lucilia sericata* Meigen (Diptera: Calliphoridae), equal 50:50 ratio, or 25% *P. regina* to 75% *L. sericata* at (**A**) 15 °C, (**B**) 25 °C, and (**C**) 35 °C across densities (larvae/jar). There was an interaction between density, species treatment, and temperature (F_3,584_ = 6.02, *p =* 0.0005).

**Figure 4 insects-14-00139-f004:**
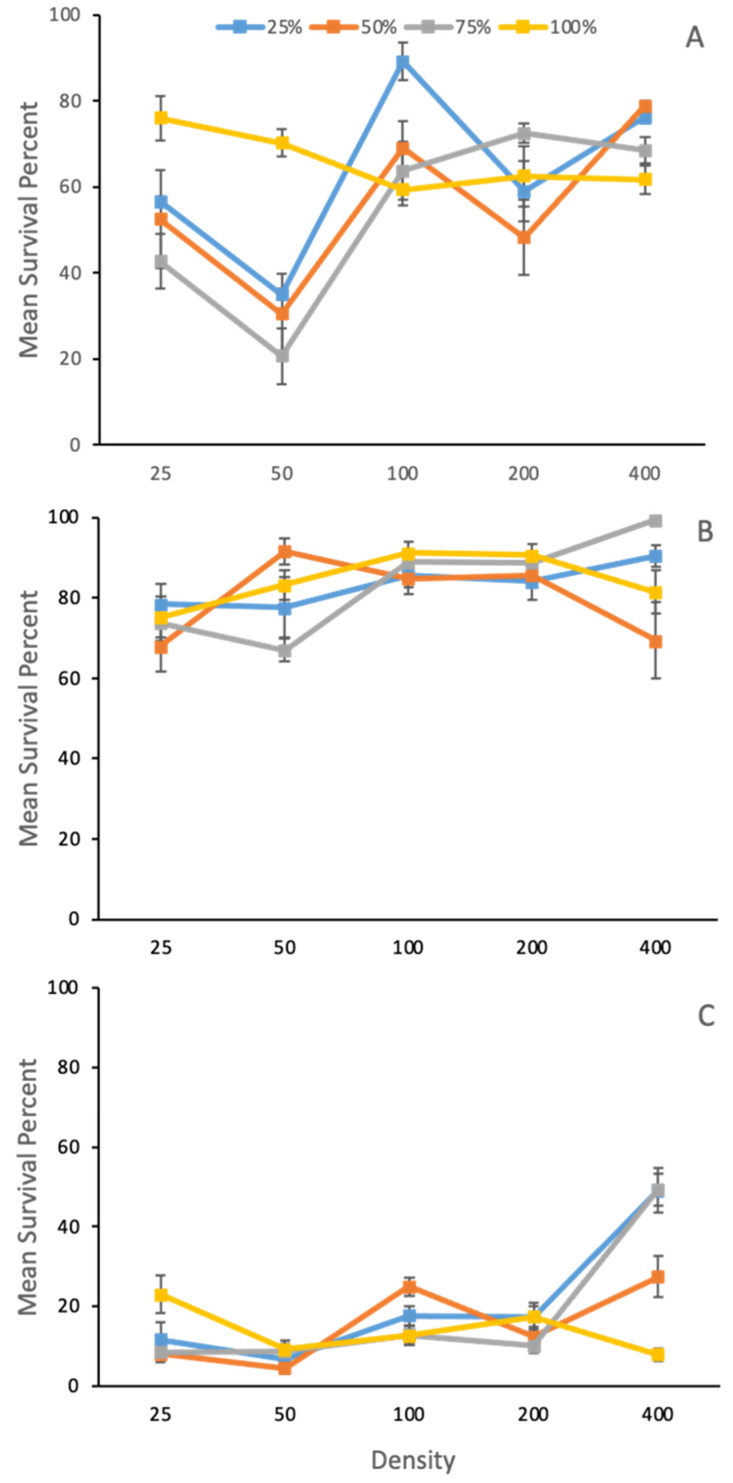
Mean ± SE percent survival to adult of *Lucilia sericata* Meigen (Diptera: Calliphoridae) reared within larval ratios of 100% *L. sericata*, 75% *L. sericata* to 25% *Phormia regina* Meigen (Diptera: Calliphoridae), equal 50:50 ratio, or 25% *L. seriata* to 75% *P. regina* at (**A**) 15 °C, (**B**) 25 °C, and (**C**) 35 °C across densities (larvae/jar). Survival was influenced by an interaction between density and species ratio treatments (F_3,584_ = 7.19, *p <* 0.0001), but not temperature (F_3,584_ = 0.77, *p =* 0.5).

**Table 1 insects-14-00139-t001:** General linear model testing the effects of temperature, density and species treatment on the female body size of *Phormia regina* Meigen and *Lucilia sericata* Meigen (Diptera: Calliphoridae). α = 0.05 for all effects.

Effects Tests for *P. regina*	DF	Sum of Squares	F Ratio	*p-Value*
Density	1	0.8369	69.148	<0.0001
Species treatment	3	4.5366	124.935	<0.0001
Temperature	1	0.1740	14.382	0.0002
Density × species treatment	3	0.6348	17.483	<0.0001
Density × temperature	1	0.0628	5.190	0.0229
Species treatment × temperature	3	0.1464	4.033	0.0073
Density × temperature × species treatment	3	0.1442	3.972	0.0079
**Effects Tests for *L. sericata***	**DF**	**Sum of Squares**	**F Ratio**	** *p-Value* **
Density	1	0.0003	0.038	0.8453
Species treatment	3	2.9960	123.229	<0.0001
Temperature	1	3.6326	448.199	<0.0001
Density × species treatment	3	0.1628	6.697	0.0002
Density × temperature	1	0.1319	16.275	<0.0001
Species treatment × temperature	3	0.2155	8.864	<0.0001
Density × temperature × species treatment	3	0.2078	8.549	<0.0001

**Table 2 insects-14-00139-t002:** Analysis of variance testing the effects of temperature, density, and species treatment survival to adult of *Phormia regina* Meigen and *Lucilia sericata* Meigen (Diptera: Calliphoridae). α = 0.05 for all effects.

Effects Tests for *P. regina*	DF	Sum of Squares	F Ratio	*p-Value*
Density	1	0.1622	2.864	0.09
Species treatment	3	1.9362	11.391	0.0001
Temperature	1	17.8180	314.477	<0.0001
Density × species treatment	3	1.0880	6.403	0.0003
Density × temperature	1	0.5189	9.160	0.0026
Species treatment × temperature	3	4.1607	24.478	<0.0001
Density × temperature × species treatment	3	1.0244	6.027	0.0005
**Effects Tests for *L. sericata***	**DF**	**Sum of Squares**	**F Ratio**	** *p-Value* **
Density	1	2.2081	30.159	<0.0001
Species treatment	3	0.0783	0.357	0.78
Temperature	1	20.0428	273.750	<0.0001
Density × species treatment	3	1.5807	7.197	<0.0001
Density × temperature	1	0.0034	0.047	0.83
Species treatment × temperature	3	0.2264	1.031	0.33
Density × temperature × species treatment	3	0.1697	0.773	0.51

## Data Availability

The data presented in this study are openly available in figshare.com at 10.6084/m9.figshare.21936480 (accessed 20 January 2023).

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
