# Peer review of "Density, Temperature, and Comingled Species Affect Fitness within Carrion Communities: Coexistence in Phormia regina and Lucilia sericata (Diptera: Calliphoridae)"

_insects, 2023, doi:10.3390/insects14020139_

Round 1
Reviewer 1 Report
The authors demonstrated complicated interaction between abiotic (temperature) and biotic (population density) factors determining larval survival and adult size in two most common blow fly species. The design of the experiment and the analysis of the data are correct. The results of the present study are certainly important for both fundamental insect ecology and applied (forensic) entomology and therefore the manuscript can be published, although it needs a number of minor corrections and improvements (see below).
Lines 14, 34, 35, 40, 41, 44, etc. – delete unnecessary hyphens
Line 34: I guess that the right order of words should be: “coexist on carrion, a temporary resource, successfully”.
Line 38: Latin names Lucilia sericata and Phormia regina should be in Italics
Line 41: P. regina should be in Italics
Lines 42, 43: L. sericata should be in Italics
Line 54: “non-interactive interactions” sounds strange. Possibly, you mean “non-interactive communities”?
Line 156: I would replace “Mean ± S.E. “ with “here and below Mean ± S.E. are given” and then I would delete “Mean ± S.E.” in lines 168 and 169.
Line 208, 221, 223,237, 239 etc: I think that species authority can be indicated only once, in the beginning of the paper
Table 1: Please align all columns in upper and lower parts of the table.
Figures 1 and 2: I would suggest starting the vertical axis not from 0 but from 0.5. This would make graphs more clear.
Lines 256-257: the degree sigh should be at the top of the number.
Line 334: The temperature-size rule was proposed much earlier than in 2003. See, for example, Atkinson, D., & Sibly, R. M. (1997). Why are organisms usually bigger in colder environments? Making sense of a life history puzzle. Trends in ecology & evolution, 12(6), 235-239.
Author Response
- The hyphens are due to the journal's template, not our manuscript as any words that don't fit are hyphenated and moved to the next line by the template.
- Line 34 has been corrected as suggested.
- Missing italics in lines 38, 41-43 have been fixed
- Agreed, the wording was awkward. It has been revised to 'interactive relationships'.
- This has been corrected as suggested.
- As figures and tables may be extracted from the original article source by subsequent users who display the figure/table in a presentation or some other format, these headings need to stand on their own in the absence of the article. Thus, species authority has not been removed from figure/table headings.
- Alignment in both Table 1 and 2 have been corrected.
- Y-axis has been changed to start at 0.5 for Figure 1 and 2 as suggested.
- Degree symbol placement has been corrected.
- Line 334 - Agreed. The citation has been corrected to use the earlier paper as suggested.
Reviewer 2 Report
Please see attached file

Author Response
The general comments were encompassed within specific comments below, thus addressed as they appear within each section.
Introduction comments:
- The sentence now ends with networks.
- The word has been replaced with relationships.
- Citation 6 was removed.
- The example was replaced by blow fly example of a temperature-dependent interspecific competition.
- Reference, order and family was added.
- Species authority was removed since Lucilia sericata had been previously introduced.
- Line 105: Citation 18 (previously 19) was updated to an insect example instead of a mite example.
- Line 125: Phormia regina has been introduced in line 82, this was as a result of including a blow fly example of temperature-dependent interspecific competition as suggested.
- Line 126-131: There is slight disagreement in the literature as to the exact upper development limit, with variation between sources cited. As we continue to study and work with these flies, it is apparent that they do not have the exact same upper temperature range.
- Line 131-133: Citations have been added that examine interactions between L. sericata and P. regina.
Methods and Materials comments:
- Line 166-167: the sentence was moved to come before the section concerning oviposition, as suggested.
- Line 167-170: We have clarified the following within the manuscript, now reflected in lines 173-180: Liver was available to the colony cages for oviposition for 4 hours. Eclosion was checked every 4-6 hours. Based on the frequency of checking for eclosion and the difference between eclosion times, larvae were all less than 4-6 hours old when placed. Typically, this reflected 2-3 hours due to variability in eclosion of L. sericata in particular, however we've modified our statement to clarify and include the maximum potential age of first instars at the time of placement. At the highest larval density, 50 g of liver was sufficient for 100% survival, albeit small individuals, hence no additional liver was provided. Furthermore, addition of more liver would have invalidated density treatments as species interactions are ameliorated when resources are plentiful. Nor was water added as all treatment conditions were maintained equally and addition of water would change humidity conditions preferentially if it wasn't added to all treatment equally.
- Line 170: No, the liver was not in a container. First instars were transferred onto the pork liver and the liver was placed directly into the mason jar on top of the wood shavings pupation media.
- Section 2.2: More information on how the replicates were randomized has been provided in lines 192-194. Three growth chambers were used, each with five shelves with randomized species ratio across shelves and density randomized within shelves.
- Section 2.2: Preliminary data tested for the presence of any elevation of temperature due to maggot masses. The highest density of 400 larvae never elevated within jar temperatures beyond the growth chamber set temperature. This has been added to the manuscript.
- Line 200-201: We've added 'female' to reflect that we present cross-vein length measurements for adult females.
Results:
- Tables 1 & 2 issues with alignment has been corrected
- Table 1: As suggested, analysis of variance was changed to general linear model.
- Figures 1- 4: As requested by reviewer 1, we changed the vertical axis for body size figures to start at 0.5 to expand the individual graphs and make the error bars clearer, however, low variability resulted in small error bars, which doesn't require further explanation as this is self-explanatory. There is no option to move the error bars in front of the point in the graphing program.
- Figures 1- 4: All data points have been converted to squares.
- Figure 1-4: As figures may be extracted from the source and used in presentations and other citations by subsequent researchers, we have left the final sentence to ensure clarity if these figures are used on their own.
- Figures 1-2: y-axis has been re-labelled as “mean wing vein”
- As suggested, these redundant sentences were removed
- Line 211-212: the exceptions to the trend were added as suggested.
- Line 232-234: We suggest that the reviewer perhaps misread as together with the previous sentence, the statement refers to conspecific treatments of L. sericata across the 3 temperatures tested.
- Line 242-244: We are unable to discern your meaning as your statement “Many times the 100% ratios were either equal to or smaller than the 100% treatment” is circular as 100% ratios and 100% treatments are the same treatments. However, we changed the statement “at the lowest density of 50” to “at a low density of 50” because survival at 25 was higher than survival at 50.
- Line 263-264: The suggested exceptions to the trend were added.
Discussion:
- We have chosen not to cite tables/figures in the discussion as it is not an acceptable style within most journal articles, however should the editor suggest this change, we would be happy to reconsider.
- We have already addressed the lack of elevated temperature at the highest density in point 5 within methods above. The observation suggested by the reviewer is also explained by the mid-temperature of 25oC being an optimal temperature, with 15 and 35oC within the species tolerance for development but not ideal.
- Line 427-433: information about the effect of geographic origin on Lucilia sericata competitive ability has been added, as suggested.
- Line 124-128: information on founder control and the relationship with species density ratios was added in the introduction, as suggested.
Citations:
- Citation #5 was updated as suggested.
- Citation #14, Line 90-92: changes to the wording were made to clarify the relationship between Tmax, Tmin, tolerance zones and optimal zones, and that it was development outside the optimal range but within the limits of Tmax and Tmin that we were discussing, as suggested.
- Lines 74, 98 &141: year and citation number were provided.
- Citation #22 has been corrected to Satar S.
Reviewer 3 Report
The authors have graphed and presented their results clearly, drawing some attention to the implications of their findings. I found the study of interest and a good contribution to the knowledge of bio ecology of blow flies. The methods used are appropriate for the objectives of the work and, in general, well depicted. The resulting figures are sufficient, informative, and of good quality helping to follow the reasoning throughout the manuscript. The discussion of results and comments on future research should be improved if the paper is to be accepted for publication in Insects.
My primary concern is that the authors are extrapolating the applicability of their results beyond what the design supports. These are only data from a set of three highly artificial constant laboratory temperature conditions ranging from 15 to 35 C, so the inference power of the paper is very limited, but authors do not acknowledge this detail at all and need to be more forthcoming. The effect of fluctuating temperature profiles on blow flies was not investigated in this study. This is a critical limitation of the study, and the authors must concede and discuss this. The interaction of cyclic temperatures with curve linear characteristics of blow flies (or insects in general) development curves can introduce significant deviations from the results obtained here, and especially at the lower and higher temperatures of development functions that were not investigated in this study. Studies across a broader set of fluctuating temperature regimes are therefore encouraged so that more realistic effect of temperature on biological parameters of blow flies could be understood, as this is the closest to temperature fluctuations that occur in the field. So, I am suggesting to the authors to tone-down the language a little and admit that there are still substantive uncertainties to be considered.
Some of the authors’ statements would be much stronger if they tie their work to the body of literature that has built up on the bio ecology of hymenopteran parasitoids, e.g., Journal of Economic Entomology 112:1560-1574 and Journal of Economic Entomology 112:1062-1072. These studies provide strong evidence that daily temperature fluctuations significantly affected development times (and longevity) of parasitoids studied, resulting in marked deviations and potentially erroneous predictions when compared to their constant temperature regimen counterparts. This article should provide details on all these fronts to provide the proper context for the work. This is not to diminish the data gathered in this study, as they are of value. But it is important for the authors not to overgeneralize, and to warn the reader, including regulatory agencies, against doing so as well.
Adding these details will improve the discussion in my humble opinion.
Good luck!
Author Response
The reviewer states that rearing conditions for this study due to use of constant laboratory temperature make these data limited in their applicability, however does not recognize that these are the very rearing conditions forensic entomologists use to develop temperature relationships used in estimating post-mortem intervals with blow fly development. There are certainly limitations to these methods, absolutely, yet discussion of these limitations could and has filled review papers. This is beyond the scope implied by the current study. Future work should absolutely include fluctuating temperatures, which we have now added to lines 477-483 in a paragraph devoted to addressing this issue, as this is certainly a limiting factor that we acknowledge.
With regard to the reviewer's comments regarding inclusion of parasitoid literature effects on fluctuating temperatures as a proxy for blow flies, we have chosen instead to include the current literature relevant to fluctuating temperatures on development of blow flies, and discuss the status of this research as it pertains to blow flies in the hopes that in addition to recognizing the limitations of our own study, we point out future rich areas of further study.
Round 2
Reviewer 3 Report
Authors have done a fine job addressing all of my original comments and concerns and those of other reviewers. Thank you.